# D-Dimers Level as a Possible Marker of Extravascular Fibrinolysis in COVID-19 Patients

**DOI:** 10.3390/jcm10010039

**Published:** 2020-12-24

**Authors:** Antonin Trimaille, Jecko Thachil, Benjamin Marchandot, Anaïs Curtiaud, Ian Leonard-Lorant, Adrien Carmona, Kensuke Matsushita, Chisato Sato, Laurent Sattler, Lelia Grunebaum, Yves Hansmann, Samira Fafi-Kremer, Laurence Jesel, Mickaël Ohana, Olivier Morel

**Affiliations:** 1Division of Cardiovascular Medicine, Nouvel Hôpital Civil, Strasbourg University Hospital, 67000 Strasbourg, France; antonin.trimaille@chru-strasbourg.fr (A.T.); benjaminmarchandot@gmail.com (B.M.); anais.curtiaud@chru-strasbourg.fr (A.C.); adrien.carmona@chru-strasbourg.fr (A.C.); matsuken_22@yahoo.co.jp (K.M.); okoge16@gmail.com (C.S.); laurence.jesel@chru-strasbourg.fr (L.J.); 2INSERM (French National Institute of Health and Medical Research), UMR 1260, Regenerative Nanomedicine, FMTS, 67000 Strasbourg, France; 3Department of Haematology, Manchester Royal Infirmary, Oxford Road, Manchester M13 9WL, UK; jecko.thachil@mft.nhs.uk; 4Department of Radiology, Nouvel Hôpital Civil, Strasbourg University Hospital, 67000 Strasbourg, France; leonard.lorant@gmail.com (I.L.-L.); mickael.ohana@chru-strasbourg.fr (M.O.); 5Laboratory of Haematology, Thrombosis and Haemostasis Unit, Strasbourg University Hospital, 67000 Strasbourg, France; laurent.sattler@chru-strasbourg.fr (L.S.); lelia.grunebaum@chru-strasbourg.fr (L.G.); 6Department of Infectious Diseases, Nouvel Hôpital Civil, Strasbourg University Hospital, 67000 Strasbourg, France; yves.hansmann@chru-strasbourg.fr; 7Department of Virology, Nouvel Hôpital Civil, Strasbourg University Hospital, 67000 Strasbourg, France; Samira.fafi-kremer@chru-strasbourg.fr

**Keywords:** coronavirus, fibrinolysis, lung injury, respiratory distress syndrome, thrombosis

## Abstract

Background and Objective: Host defence mechanisms to counter virus infection include the activation of the broncho-alveolar haemostasis. Fibrin degradation products secondary to extravascular fibrin breakdown could contribute to the marked increase in D-Dimers during COVID-19. We sought to examine the prognostic value on lung injury of D-Dimers in non-critically ill COVID-19 patients without thrombotic events. Methods: This study retrospectively analysed hospitalized COVID-19 patients classified according to a D-Dimers threshold following the COVID-19 associated haemostatic abnormalities (CAHA) classification at baseline and at peak (Stage 1: D-Dimers less than three-fold above normal; Stage 2: D-Dimers three- to six-fold above normal; Stage 3: D-Dimers six-fold above normal). The primary endpoint was the occurrence of critical lung injuries on chest computed tomography. The secondary outcome was the composite of in-hospital death or transfer to the intensive care unit (ICU). Results: Among the 123 patients included, critical lung injuries were evidenced in 8 (11.9%) patients in Stage 1, 6 (20%) in Stage 2 and 15 (57.7%) in Stage 3 (*p* = 0.001). D-Dimers staging at peak was an independent predictor of critical lung injuries regardless of the inflammatory burden assessed by CRP levels (OR 2.70, 95% CI (1.50–4.86); *p* < 0.001) and was significantly associated with increased in-hospital death or ICU transfer (14.9 % in Stage 1, 50.0% in Stage 2 and 57.7% in Stage 3 (*p* < 0.001)). D-Dimers staging at peak was an independent predictor of in-hospital death or ICU transfer (OR 2.50, CI 95% (1.27–4.93); *p* = 0.008). Conclusions: In the absence of overt thrombotic events, D-Dimers quantification is a relevant marker of critical lung injuries and dismal patient outcome.

## 1. Introduction

In December 2019, the worldwide medical community discovered that a virus called severe acute respiratory syndrome coronavirus 2 (SARS-CoV-2) could affect its ability to ensure the care and safety of patients. Modelling the course of SARS-CoV-2 infection is particularly challenging as no scoring system currently exists to guide hospital admission and treatments. Before the coronavirus disease 2019 (COVID-19) era, the CURB-65 and the Pneumonia Severity Index (PSI) scores were the two leading scoring systems used to assess the illness severity of community acquired pneumonia [1,2]. With the emergence of COVID-19, the decision-making process is undermined.

Measurement of D-Dimers, products of fibrin degradation, has proven useful to evaluate patient severity with community-acquired pneumonia [3,4], 2009 novel influenza A (H1N1) [5] and other members of the *Coronaviridae* family [6]. SARS-CoV-2 infection is associated with an increased risk of venous thrombo-embolic events [7,8,9,10,11,12], and COVID-19 coagulation abnormalities are recognized as a major determinant of dismal prognosis [13]. D-Dimers are consistently and markedly raised in patients with COVID-19 [13,14,15,16,17]. Most publications, to date, have attributed this increase to coagulation activation, and D-Dimers have been viewed as a marker for clot breakdown [9,13]. On the basis of this assumption, several clinicians have considered using D-Dimers as a surrogate marker for anticoagulant intensification [8,18]. However, in this respect they may be overlooking the non-thrombotic D-Dimers, and the intensified anticoagulation regimen may not be required. D-Dimers may signify extra-vascular fibrin degradation, in addition to fibrinolytic breakdown of the pulmonary thrombi [19]. Since the extravascular D-Dimers may be generated by the breakdown of fibrin, which is generated from fibrinogen that leaked out of the intravascular space [19], it could also be a marker of lung oedema.

In this study, we assess whether plasmatic levels of D-Dimers, as a reflection of extravascular fibrinolysis, correlated with illness severity and lung injuries in COVID-19 patients hospitalized in general wards without venous thrombo-embolic events.

## 2. Methods

### 2.1. Study Population

We identified consecutive COVID-19 patients with available D-Dimers measurements admitted in nine general departments at Strasbourg University Hospital (two centres: Nouvel Hôpital Civil (NHC) and CHU Hautepierre) from February 25 to April 19, 2020 (Figure 1). COVID-19 was confirmed by a positive result of a reverse-transcriptase–polymerase-chain-reaction (RT-PCR) assay of a specimen collected on a nasopharyngeal swab according to the World Health Organization guidance [20]. Patients with typical findings of COVID-19 at chest computed tomography (CT; i.e., bilateral and peripheral ground glass opacities or alveolar consolidations), and for whom testing for the COVID-19 virus was either inconclusive or could not be performed, were considered as confirmed COVID-19 cases by a multidisciplinary team. Medical management was left at the discretion of the treating physician. The Strasbourg University Hospital COVID-19 Clinical Guidelines did not recommend a systematic and specific thromboprophylaxis in ward patients at the time of the present study. When decided, thromboprophylaxis was achieved with Enoxaparin at 4000 international units (IU)/24 h, Fondaparinux at 2.5 mg/24 h or unfractionated heparin at 4800 IU/24 h (standard preventive treatment), with Enoxaparin at 4000 IU twice/24 h (reinforced preventive treatment) or with Enoxaparin 100 IU/kg twice/24 h (therapeutic).

### 2.2. Data Collection

Anonymized patient data were retrospectively reviewed from the day of admission until transfer to the intensive care unit (ICU), in-hospital death or discharge from hospital. Data included demographics, clinical characteristics, comorbidities, symptoms, medication and laboratory and radiological findings. The D-Dimers kinetic was defined as the difference between the peak value during hospitalization and the admission value of D-Dimers. Laboratory and imaging testing were performed according to clinical care needs. D-dimers were measured on STA-R^®^ Evolution (Diagnostica Stago^®^, Asnières-sur-Seine, France) with standard commercial reagents and protocols.

This study was approved by the research ethics committee of Strasbourg Hospital (authorization CE-2020-57) and waived the need of informed consent.

### 2.3. Study Definitions

Patients were categorized into three stages (independent, not additive) depending on the D-Dimers level according to the proposed COVID-19 associated haemostatic abnormalities (CAHA) classification [21]: Stage 1 for D-Dimers less than three-fold above normal, Stage 2 for D-Dimers three- to six-fold above normal and Stage 3 for D-Dimers greater than six-fold above normal. Lesion severity on CT performed at admission was classified following the ESR/ESTI guidelines [22], assessed as pulmonary injuries extension in percentage of the total lung parenchyma and classified as minimal (<10%), moderate (10 to 25%), severe (25 to 50%) or critical (>50%). Patients with overt thrombotic events during hospitalization, including acute pulmonary embolism (APE), deep vein thrombosis (DVT) and/or cerebral venous thrombosis (CVT), were excluded from the analysis to avoid any confusion in the D-Dimers level interpretation. Patients without D-Dimers kinetics were excluded from the analysis.

### 2.4. Study Outcomes

The primary outcome was the occurrence of critical pulmonary injuries on chest CT (pulmonary lesions extension >50% of the total lung parenchyma). The secondary outcome was the composite of in-hospital death or transfer to the ICU.

### 2.5. Statistical Analysis

Continuous variables were expressed as mean ± standard deviation or median (interquartile range) and categorical variables as counts and percentages. Continuous variables were compared with the use of parametric (ANOVA) or non-parametric Mann–Whitney tests as appropriate. Categorical variables were compared with chi-square test or Fischer’s exact test. Associations of potential prognosis factors and the risk of critical pulmonary injuries extension were first investigated in univariable analysis. Factors associated with a *p* value < 0.05 were entered into a logistic regression model.

Statistical analyses were performed using SPSS 17.0 for Windows (SPSS Inc., Chicago, IL, USA).

## 3. Results

### 3.1. Characteristics

In total, 123 consecutive patients with confirmed COVID-19 infection, available D-Dimers kinetics and an absence of overt thrombotic events were admitted to general wards between February 25 and April 19, 2020 (Figure 1). The prevalence of D-Dimers stages is presented in Table 1. According to this staging scheme, 67 (54.5%) patients were in Stage 1, 30 (24.4%) were in Stage 2 and 26 (21.1%) were in Stage 3. Patients in Stage 3 were more likely to be male than in either other stage (52.2% were male in Stage 1, 53.3% in Stage 2 and 80.8% in Stage 3, *p* = 0.035) and to suffer from heart failure (*p* = 0.004). By contrast, they were less likely to have a previous history of chronic obstructive pulmonary disease (COPD; *p* = 0.017) (Table 1). Standard thromboprophylaxis was less frequently used in Stage 3 patients (*p* = 0.047). C-Reactive protein (CRP) on admission and at peak, fibrinogen at peak and B-type natriuretic peptide (BNP) on admission were at their highest levels in Stage 3 patients (*p* < 0.05 for all). In addition, kidney injuries were more pronounced in Stage 3 patients (Table 2).

### 3.2. Chest CT Findings, Lactates and Oxygen Demand

The extent of COVID-19 disease assessed by chest CT is presented in Figure 2 and Appendix A. A stepwise increase in critical lung injuries that paralleled D-Dimers stages was evidenced (11.9% of critical lung injuries in Stage 1, 20.0% in Stage 2 and 57.7% in Stage 3, *p* = 0.001). Likewise, maximum oxygen flow rates and lactate levels were at their highest levels in Stage 3 patients (*p* < 0.05 for both) (Table 2 and Table 3).

### 3.3. Predictors of Critical Pulmonary Injuries

In univariable analysis, the following factors were associated with the occurrence of critical lung injuries on chest CT: fibrinogen on admission, CRP on admission or at peak, lymphocytes on admission and D-Dimers staging on admission or at peak. By multivariable analysis, D-Dimers staging at peak remained the sole independent predictor of critical lung injuries regardless of the inflammatory burden (odds ratio (OR) 2.70, 95% confidence interval (CI) 1.50–4.86, *p* < 0.001) (Table 4).

### 3.4. Outcomes

The composite of in-hospital deaths or transfer to the ICU occurred in 40 (32.5%) patients in the overall population. According to D-Dimers staging at peak, its incidence was 14.9% in Stage 1, 50.0% in Stage 2 and 57.7% in Stage 3 (*p* < 0.001) (Table 3). According to D-Dimers staging at admission, its incidence was 27.8% in Stage 1, 52.2% in Stage 2 and 31.2% in Stage 3 (*p* = 0.091) (Appendix A).

In the univariable analysis, male sex, Fibrinogen at peak, CRP at peak and D-dimers staging at peak were associated with in-hospital death or ICU transfer. According to the multivariable analysis, D-dimers staging at peak remained the sole independent predictor of adverse prognosis (OR 2.50, 95% CI95% 1.27–4.93, *p* = 0.008) (Appendix A).

## 4. Discussion

This study was specifically designed to assess the prognosis impact of D-Dimers as a possible marker of extravascular fibrinolysis in COVID-19 patients without overt thrombotic events. We showed a stepwise increase in critical lung injuries according to D-Dimers staging following the CAHA classification at baseline or at peak. D-Dimers staging was an independent predictor of critical lung injuries and adverse prognosis regardless of the extent of inflammatory burden. These findings suggest that this staging system could be an additive tool to enhance risk stratification and therapeutic decision making in COVID-19 patients hospitalized out of the ICU even in the absence of clot formation.

### 4.1. Extravascular Fibrinolysis during COVID-19

The role of abnormal coagulation parameters was quickly emphasized during the COVID-19 pandemic, especially that of raised D-Dimers [13,14]. In early reports, these modifications were attributed to disseminated intravascular coagulation or sepsis-induced coagulopathy [13,23]. While the frequency of venous thrombotic events during COVID-19 is increased and correlated with worse outcomes [10,24], D-Dimers can be elevated even in patients without evidence of thrombosis [25]. Being the end-products of fibrin degradation in both intravascular and extravascular sectors, D-Dimers measurement is a well-recognized rule-out diagnostic tool for venous thromboembolism. Here, we hypothesized that COVID-19 is associated with an increase in extravascular fibrinolysis, correlated with illness severity, and that D-Dimers could reflect this phenomenon. We used a D-Dimers level staging following the proposed CAHA classification [21] to assess its prognostic values during COVID-19. This staging at baseline or at peak during hospitalization is associated with an increase in lung injuries assessed by CT scans in patients hospitalized in general wards without evidence of venous thrombotic events, and it was an independent predictor of worse in-hospital outcomes.

Similar to severe acute respiratory syndrome coronavirus (SARS), SARS-CoV-2 infects host cells through its functional receptor, angiotensin converting enzyme 2 (ACE2), which is an aminopeptidase exposed at the extracellular surface of lung alveolar epithelial cells and endothelial cells [26]. Endotheliitis secondary to endothelial cells infection [27] and the diffuse alveolar and pulmonary interstitial inflammation secondary to lung cells invasion together drive the extensive immuno-thrombosis in the pulmonary vascular bed [28]. In addition to the pulmonary intravascular coagulopathy, our results suggest that COVID-19 leads to a pulmonary extravascular fibrinolysis that is proportional to the illness severity and reflected by D-Dimers levels. We used a specific design that excluded patients with established VTE in order to avoid any confusion in the D-Dimers level interpretation between intravascular fibrinolysis that is secondary to venous thrombosis and extravascular fibrinolysis that follows SARS-CoV-2 infection.

The presence of intra-alveolar fibrin in patients with acute lung injury (ALI) has been known for more than a decade [29]. It has been suggested that this may be a physiological process in the beginning since it may interestingly aid in alveolar gas exchange by sealing leakage sites, in addition to providing a matrix for wound repair and tissue remodelling [30,31]. However, if it becomes marked and persistent, intra-alveolar fibrin deposition can lead to irreversible changes of the hyaline membrane and fibrosis [30]. Lung surfactant can become incorporated into the fibrin network resulting in loss of lung elasticity [31]. In a study of broncho-alveolar lavages from patients who developed acute respiratory distress syndrome (ARDS), D-Dimers in the lavage fluid showed positive correlation with its protein content, which suggests much of these D-Dimers may be coming from extravascular fibrin breakdown [29].

In the case of lung infection, the broncho-alveolar haemostasis is a lung-specific coagulation system which contributes to countering the effects of virus invasion [32]. Early steps of viral invasion in the lung comprise airway inflammation and the leakage of various plasma proteins, including thrombin and fibrinogen. In healthy individuals, the high fibrinolytic activity drives fibrin cleavage and allows the clearance of fibrin deposited in the alveolar compartment and the continuation of gas exchange [33]. However, persistence of extravascular fibrin might lead to a vicious circle involving inflammation and haemostasis. On one hand, this extravascular fibrin network represents a matrix for inflammatory cells to attach to and function in. On the other hand, during severe COVID-19 lung inflammation, plasmin or proteolytic enzymes released from activated neutrophils contribute to extravascular fibrin breakdown [34]. The subsequent generation of D-Dimers, even in the absence of clot formation in the vasculature, is a plasmatic translation of this extravascular fibrinolysis. This interplay between inflammation and fibrinolysis is highlighted in our study by a stepwise increase in CRP at admission and at peak following the D-Dimers staging.

We did not observe any difference in the use of reinforced thromboprophylaxis or therapeutic anticoagulation in the different stages of D-Dimers classification. Standard thromboprophylaxis was used less frequently in Stage 3. Of note, a significant proportion of patients were under therapeutic anticoagulation even without any VTE. This finding probably results from the lack of clinical guidelines at the time of the study and the concern of clinicians to limit thrombotic burden, especially in patients with the highest D-Dimers levels.

Interestingly, none of the patients in our study had a history of COPD among those in stages 2 or 3. COPD is characterized by a progressive airflow limitation caused by a chronic lung injury secondary to different mechanisms including inflammation, oedema and alveolar secretions [35]. Chronic destruction of pulmonary parenchyma during COPD might lead to an impaired extravascular fibrinolysis, decreasing the host’s defence against SARS-CoV-2 and explaining worse outcomes in COVID-19 patients with a prior COPD [36,37].

### 4.2. Clinical Implications

COVID-19 is an infectious model of ALI, and our results could pave the way for more widespread use of D-Dimers as a marker of extent and magnitude of lung injuries. Development of lung oedema is a common feature of many conditions like sepsis and trauma, which has ARDS as the final outcome [38]. It would be useful to identify the early onset of ARDS in these scenarios to plan better supportive care and interventions. D-Dimers represent a candidate biomarker to suggest the onset and severity of pulmonary leakage, especially in the absence of thrombi formation, and improve non-invasive monitoring of ARDS. Further studies are needed to confirm the role of D-Dimers monitoring for risk assessment in other lung pathologies.

### 4.3. Study Limitations

Our study has some limitations. First, the retrospective character of the data collection should be emphasized. Second, this work was conducted in Strasbourg University Hospital, located in one of the largest COVID-19 clusters in France. The high rate of disease transmission in the region has potentially increased COVID-19 severity. Third, since there was no systematic screening of VTE, we cannot exclude that there were patients included in the analysis with asymptomatic VTE. While computed tomography pulmonary angiography was systematically performed whenever APE was suspected, complete duplex ultrasound for deep vein thrombosis diagnosis was used sparingly, and an under-estimation of deep vein thrombosis in the overall cohort is possible. However, more than one–third of the population have benefited from a VTE imaging test (100 patients (34.6%)). Fourth, we cannot exclude the possibility of confounding in line with different therapeutic doses of thromboprophylaxis. Finally, while the D-Dimers level may be an interesting biomarker to reflect the severity of lung injury during COVID-19, limitations on the measurement of the plasma concentration of D-dimers may exist, depending on the methodology used and translated by a large inter-laboratory variability [39]. Tests performance seems to be suboptimal, in particular for the highest values. This point justifies the use of ratios in comparison to the upper limit of the normal value of D-Dimers.

## 5. Conclusions

In the absence of overt thrombotic events, D-Dimers staging is a possible indicator of extravascular fibrinolysis and thus a relevant marker of critical lung injuries and dismal outcome in COVID-19 patients.

## Figures and Tables

**Figure 1 jcm-10-00039-f001:**
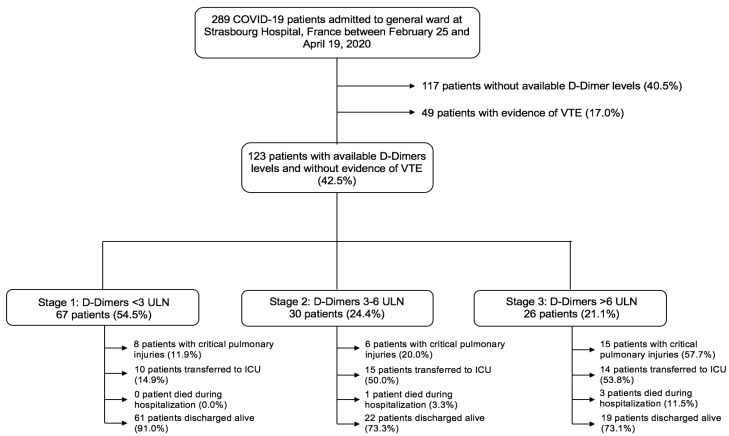
Flow-chart of the study. Of the 289 COVID-19 patients admitted to general ward at Strasbourg Hospital, France, between 25 February and 19 April 2020, this study included 123 patients, after we excluded 117 patients who did not have available D-Dimers levels and 49 patients with venous thrombo-embolism during hospitalization. According to a D-Dimers staging following the CAHA classification, 67 patients (54.5%) were in Stage 1, 30 (24.4%) in Stage 2 and 26 (21.1%) in Stage 3. Abbreviations: COVID-19, coronavirus disease 2019; ICU, intensive care unit; VTE, venous thromboembolism; ULN, upper limit of normal.

**Figure 2 jcm-10-00039-f002:**
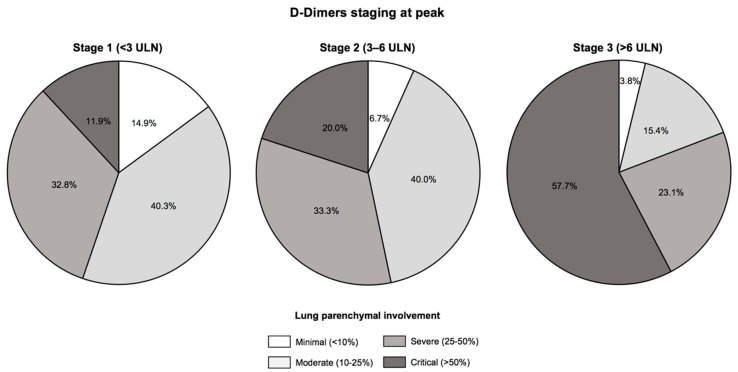
Extent of lung injuries assessed by chest computed tomography according to D-Dimers staging at peak. There was a stepwise increase in lung injuries according to D-Dimers staging at peak, with critical lung injuries defined by an extent >50% on chest computed tomography evidenced in 8 (11.9%) patients in Stage 1, 6 (20.0%) in Stage 2 and 15 (57.7%) in Stage 3 (*p* = 0.001). Abbreviations: ULN, upper limit of normal.

**Table 1 jcm-10-00039-t001:** Baseline clinical characteristics according to D-Dimers staging at peak.

	D-Dimers Staging at Peak			*p* Value
	Stage 1 <3 ULN (*n* = 67)	Stage 2 3–6 ULN (*n* = 30)	Stage 3 >6 ULN (*n* = 26)	
Age—yr	57 ± 16	62 ± 16	63 ± 16	0.120
Male—*n* (%)	35 (52.2)	16 (53.3)	21 (80.8)	**0.035**
BMI > 30 kg/m^2^—*n* (%)	21 (35.6)	12 (44.4)	7 (26.9)	0.412
Hypertension—*n* (%)	23 (34.3)	17 (56.7)	12 (46.2)	0.109
Diabetes—*n* (%)	9 (13.4)	7 (23.3)	3 (11.5)	0.379
Dyslipidemia—*n* (%)	14 (20.9)	8 (26.7)	10 (38.5)	0.222
Smoking—*n* (%)	3 (4.5)	0 (0.0)	1 (3.8)	0.507
Heart Failure—*n* (%)	1 (1.5)	0 (0.0)	4 (15.4)	**0.004**
CKD ^a^—*n* (%)	2 (3.0)	2 (6.7)	4 (15.4)	0.094
COPD—*n* (%)	9 (13.4)	0 (0.0)	0 (0.0)	**0.017**
Active cancer—*n* (%)	1 (1.5)	2 (6.7)	0 (0.0)	0.206
Thromboprophylaxis—*n* (%)	66 (98.5)	30 (100.0)	25 (96.2)	0.521
Standard—*n* (%)	41 (61.2)	21 (70)	10 (38.5)	**0.047**
Reinforced—*n* (%)	12 (17.9)	5 (16.7)	7 (26.9)	0.556
Therapeutic—*n* (%)	13 (19.4)	4 (13.3)	8 (30.8)	0.260

Data are presented as mean ± standard deviation in case of any other indication. ^a^ CKD is defined as an eGFR < 60 mL/min/1.73 m^2^. Abbreviations: BMI, body mass index; CKD, chronic kidney disease; COPD, chronic obstructive pulmonary disease; ULN, upper limit of normal. Bold *p* Value, highlight the significant *p* values.

**Table 2 jcm-10-00039-t002:** Laboratory findings at admission and during hospitalization according to D-Dimers staging at peak.

	D-Dimers Staging at Peak			*p* Value
	Stage 1 <3 ULN (*n* = 67)	Stage 2 3–6 ULN (*n* = 30)	Stage 3 >6 ULN (*n* = 26)	
**At admission**				
Lactate—mmol/L	0.95 ± 0.29	1.22 ± 0.73	1.39 ± 0.67	**0.010**
Leukocytes—×10^9^ per L	6.87 ± 3.27	6.87 ± 2.37	7.63 ± 3.70	0.563
Lymphocytes—×10^9^ per L	1.13 ± 0.50	0.92 ± 0.31	0.81 ± 0.39	**0.003**
Platelet—×10^9^ per L	224 ± 104	203 ± 81	229 ± 89	0.539
CRP—mg/L	89 ± 78	104 ± 62	152 ± 92	**0.003**
Fibrinogen—g/L	6.2 ± 1.8	6.5 ± 1.3	6.6 ± 1.1	0.636
aPTT—IU	1.1 ± 0.2	1.0 ± 0.1	1.3 ± 0.7	0.208
eGFR—mL/min/1.73m^2^	90 ± 23	70 ± 26	67 ± 31	**0.009**
BNP—pg/mL	75 ± 123	78 ± 106	669 ± 1253	**0.001**
**During Hospitalization**				
Leukocytes peak—×10^9^ per L	11.30 ± 25.12	11.48 ± 5.74	13.83 ± 8.48	0.843
Platelets peak—×10^9^ per L	399 ± 190	401 ± 137	415 ± 157	0.923
CRP peak—mg/L	125 ± 88	155 ± 88	202 ± 91	**0.001**
Fibrinogen peak—g/L	6.6 ± 1.6	7.5 ± 1.7	8.1 ± 2.6	**0.036**

Data are presented as mean ± standard deviation. Abbreviations: aPTT, activated partial thromboplastin time ratio; BNP, B-type natriuretic peptide; CRP, C-reactive protein; eGFR, estimated glomerular filtration rate; ULN, upper limit of normal. Bold *p* Value, highlight the significant *p* values.

**Table 3 jcm-10-00039-t003:** Outcomes of the study population according to D-Dimers staging at peak.

	D-Dimers Staging at Peak			*p* Value
	Stage 1 <3 ULN (*n* = 67)	Stage 2: 3–6 ULN (*n* = 30)	Stage 3 >6 ULN (*n* = 26)	
Critical pulmonary injuries (>50%)—*n* (%)	8 (11.9)	6 (20.0)	15 (57.7)	**<0.001**
Transfer to ICU or in-hospital death—*n* (%)	10 (14.9)	15 (50.0)	15 (57.7)	**<0.001**
ICU Transfer—*n* (%)	10 (14.9)	14 (46.7)	14 (53.8)	**<0.001**
In-hospital death—*n* (%)	0 (0.0)	1 (3.3)	3 (11.5)	**0.019**
Maximal oxygen flow rate—L/min	0.8 ± 0.3	2.1 ± 0.4	7.4 ± 4.6	**<0.001**
Discharged alive—*n* (%)	61 (98.4)	22 (84.6)	19 (82.6)	**0.018**

Data are presented as mean ± standard deviation in case of any other indication. Abbreviations: ICU, intensive care unit; ULN, upper limit of normal. Bold *p* Value, highlight the significant *p* values.

**Table 4 jcm-10-00039-t004:** Predictive factors of critical lung injury in COVID-19 patients.

	Univariable Analysis		Multivariable Analysis	
	OR [95%CI]	*p* Value	OR [95%CI]	*p* Value
Age	1.00 [0.98–1.02]	0.563		
Male	1.59 [0.78–3.24]	0.193		
BNP at admission	1.00 [0.99–1.00]	0.940		
Fibrinogen at admission	1.46 [1.03–2.06]	**0.030**		
Fibrinogen at peak	1.22 [0.96–1.55]	0.095		
CRP at admission	1.01 [1.00–1.01]	**<0.001**		
CRP at peak	1.00 [1.00–1.01]	**<0.001**	1.00 [0.99–1.00]	0.288
Leukocytes at admission	1.06 [0.96–1.18]	0.222		
Leukocytes at peak	1.00 [0.98–1.02]	0.503		
Lymphocytes at admission	0.38 [0.15–0.92]	**0.033**	0.79 [0.24–2.52]	0.694
Procalcitonin at admission	0.96 [0.84–1.09]	0.582		
D-Dimers staging on admission	2.37 [1.35–4.16]	**0.003**		
D-Dimers staging at peak	3.16 [1.82–5.48]	**<0.001**	2.70 [1.50–4.86]	**<0.001**

Abbreviations: BNP, B-type natriuretic peptide; CI, confidence interval; CRP, C-reactive protein; OR, odds ratio. Bold *p* Value, highlight the significant *p* values.

## Data Availability

Data sharing is not applicable to this article.

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
