# Peer review of "D-Dimers Level as a Possible Marker of Extravascular Fibrinolysis in COVID-19 Patients"

_jcm, 2020, doi:10.3390/jcm10010039_

Round 1
Reviewer 1 Report
This obervational study in COVID-19 patients without thrombosis relates Ddimer levels to the severity of pulmonary lesions, suggesting Ddimer reflects intrapulmonary fibrinolysis and is elevated in the blood due to lung edema and not due to enhanced coagulation.
I fully concur with authors that this is a possible hypothesis. It is an important consideration because it may mean that Ddimers should not guide anticoagulant prophylaxis/treatment. However, proof of this thought is not provided as this study is merely associative and this should be clear.
Some comments:
- if your hypothesis holds, how do you explain that Ddimers are a very good marker of thrombosis in COVID? Multiple papers have shown this
- do you think it is of added value to compare findings between patients with high and low intensity of anticoagulant prophylaxis? If there is no difference, this may support your hypothesis
- you provide an association between CT defined lung injury and ddimer levels, not between pulmonary leakage and ddimer levels. Do you have blood samples in which you can measure a marker of endothelial permeability?
- minor: in the abstract you describe 3 stages of lung injury, in the methods there are 4
- minor: what was the timing of CT scan imaging, and did you take the ddimer level around the same day of imaging?
- minor: in the abstract, can you add for what, when you indicate that ddimer has prognostic value
Author Response
This observational study in COVID-19 patients without thrombosis relates Ddimer levels to the severity of pulmonary lesions, suggesting Ddimer reflects intrapulmonary fibrinolysis and is elevated in the blood due to lung edema and not due to enhanced coagulation.
I fully concur with authors that this is a possible hypothesis. It is an important consideration because it may mean that Ddimers should not guide anticoagulant prophylaxis/treatment. However, proof of this thought is not provided as this study is merely associative and this should be clear.
We would like to thank the Reviewer 1 for her/his appreciation of our work. Below are all the responses to her/his relevant queries.
Some comments:
If your hypothesis holds, how do you explain that Ddimers are a very good marker of thrombosis in COVID? Multiple papers have shown this
In our study, we included in the analysis only patients without established venous thromboembolic events (VTE). We showed an increase in critical lung injuries according to D-Dimers elevation reflecting a potential implication of extravascular fibrinolysis.
As previously showed by Hoffmann et al in Cell, SARS-CoV-2 infects host cells through its functional receptor ACE2 exposed at the extracellular surface of lung alveolar epithelial cells and endothelial cells. In response, the broncho-alveolar haemostasis is activated as a specific system contributing to counter virus invasion in the lung. The subsequent activation of extravascular fibrinolysis is reflecting by the stepwise increase of D-Dimers following lung injury classification. Through a pathophysiological continuum, it may exist an endotheliitis and a diffuse alveolar and pulmonary interstitial inflammation secondary to lung cells invasion. Altogether, they drive an extensive immunothrombosis in the pulmonary vascular bed. This thrombosis burden in COVID-19 seems to be proportional to D-Dimers levels, as previously described (for instance Choi et al, Thromb Res 2020;196:318-321).
In the present study, we excluded patients with an established VTE during the hospitalization to avoid any confusion in the D-Dimers level interpretation between intravascular fibrinolysis secondary to venous thrombosis and extravascular fibrinolysis following SARS-CoV-2 infection.
To highlight this important point, we have added the following sentence in the “Discussion” section (lines 235-238): “We used a specific design with exclusion of patients with established VTE to avoid any confusion in the D-Dimers level interpretation between intravascular fibrinolysis secondary to venous thrombosis and extravascular fibrinolysis following SARS-CoV-2 infection.”
Do you think it is of added value to compare findings between patients with high and low intensity of anticoagulant prophylaxis? If there is no difference, this may support your hypothesis
We would like to thank the Reviewer 1 for this very relevant suggestion. We have added this analysis in the Table 1. Standard thromboprophylaxis was less frequently used in Stage 3 patients. This data probably underlines the concern of the clinicians to limit thrombotic burden by given reinforced or therapeutic thromboprophylaxis even in the absence of overt or documented thrombotic events. We have added the following sentences in the main text:
- “Methods” section (Lines 89-94): “The Strasbourg University Hospital COVID-19 Clinical Guidelines did not recommend a systematic and specific thromboprophylaxis in ward patients at the time of the present study. When decided, thromboprophylaxis was achieved with Enoxaparin at 4000 international units (IU)/24 hours, Fondaparinux at 2.5 mg/24 hours or unfractionated heparin at 4800 IU/24 hours (standard preventive treatment), with Enoxaparin at 4000 IU twice /24 hours (reinforced preventive treatment) or with Enoxaparin 100 IU/kg twice /24 hours (therapeutic)”
- “Results” section (Lines 145-146): “Standard thromboprophylaxis was less frequently used in Stage 3 patients (p=0.047)”
- “Discussion” section (Lines 264-269): “We did not observe any difference in the use of reinforced thromboprophylaxis or therapeutic anticoagulation in the different stages of D-Dimers classification. Standard thromboprophylaxis was less frequently used in Stage 3. Of note, a significant proportion of patients were under therapeutic anticoagulation even without any VTE. This finding probably translates the lack of clinical guidelines at the time of the study and the concern of clinicians to limit thrombotic burden, especially in patients with highest D-Dimers levels”.
You provide an association between CT defined lung injury and ddimer levels, not between pulmonary leakage and ddimer levels. Do you have blood samples in which you can measure a marker of endothelial permeability?
We would like to thank the Reviewer for this relevant proposal. Unfortunately, as described in the “Methods” section (Lines 101-102), “Laboratory and imaging testing were performed according to clinical care needs” and we did not have any sample to measure marker of endothelial permeability.
Minor: in the abstract you describe 3 stages of lung injury, in the methods there are 4
We used a D-Dimers levels classification with 3 stages (Stage 1: D-Dimers less than three-fold above normal, Stage 2: D-Dimers three-to-six-fold above normal, Stage 3: D-Dimers six-fold above normal) and a lung injury classification with 4 stages (minimal (<10%), moderate (10 to 25%), severe (25 to 50%) and critical (>50%)).
In the Abstract and the “Methods” section, we described the D-Dimers staging. The lung injury staging is reported in the “Methods” section only.
Minor: what was the timing of CT scan imaging, and did you take the ddimer level around the same day of imaging?
CT scan was performed at admission. D-Dimers were measured at admission and during hospitalization. We used both D-Dimers levels at admission (Table S2) and at peak (Tables 1,2, 3) for the staging.
To clarify, we have added the following sentence in the “Methods” section (Lines 113-115): “Lesions severity at CT performed at admission was classified following the ESR/ESTI guidelines [22] assessed as pulmonary injuries extension in percentage of the total lung parenchyma and classified as minimal (<10%), moderate (10 to 25%), severe (25 to 50%) and critical (>50%).”
Minor: in the abstract, can you add for what, when you indicate that ddimer has prognostic value
We thank the Reviewer for this suggestion. We have amended the sentence of the Abstract as follows (Lines 33-34): “We sought to examine the prognostic value on lung injury of D-Dimers in non-critically ill COVID-19 patients without thrombotic events”.
Reviewer 2 Report
Very interesting paper
L43regardless of the inflammatory burden ? please define inflammatory burden in the abstract: CRP level?
L81: with available D-Dimers measurements à how many missing data? (at least the authors should refer to the flow chart at this point)
L81: with available D-Dimers measurements the authors should mention the method and discussion the importance of the method. The authors should refer to Hardy M et al. Thomb J. 2020 Sep 7;18:17 (where this point is addressed in detail). This point also explains why authors prefer to use ratios in comparison to the upper limit of the normal range.
“Therefore, their performance at higher values, such as those proposed for initiating high dose anticoagulation in COVID-19 patients (i.e., over 3000 ng/mL), is probably suboptimal and would need to be assessed in order to avoid the use of inaccurate results in these patients. For example, external quality controls performed with moderately elevated D-dimers samples (target value 4000 ng/mL FEU) identified method-specific D-dimers means ranging from 470 to 10,150 ng/mL FEU (all methods coefficient of variation: 23%) [68]. As a possible solution, the threshold values of plasma D-dimers could be adjusted based on assay methodology. For example, for DIC diagnosis according to the ISTH definition, the appropriate D-dimer cut-off value for 2 points ranged from 3500 ng/mL to 6500 ng/mL, depending on the reagents used [69].”
L90-92: “Thromboprophylaxis was achieved with Enoxaparin at 0.4 mL per day, Fondaparinux at 2.5 mg per day or unfractionated heparin at 200 IU per hour (standard preventive treatment) or with Enoxaparin at 0.4 mL twice per day (reinforced preventive treatment)”
For LMWH and UFH:please use IU/24h or IU/kg/24h
How many patients in each group?
L165 why abbreviations appear here?
In the discussion, the authors should discuss the limitations of D-dimers see previous reference
Author Response
Very interesting paper
We would like to thank the Reviewer 2 for her/his positive appreciation and her/his in-depth review of our work. Please find below our responses to these important questions.
L43: regardless of the inflammatory burden? please define inflammatory burden in the abstract: CRP level?
We confirm that the inflammatory burden was assessed by CRP levels in our work. We have added this point in the Abstract (Lines 42-45): “D-Dimers staging at peak was an independent predictor of critical lung injuries regardless of the inflammatory burden assessed by CRP levels (OR 2.70, 95% CI (1.50-4.86); p<0.001) and was significantly associated with increased in-hospital death or ICU transfer (14.9 % in Stage 1, 50.0% in Stage 2 and 57.7% in Stage 3 (p<0.001))”.
L81: with available D-Dimers measurements à how many missing data? (at least the authors should refer to the flow chart at this point)
We have added the reference to the flow chart which shows the number of missing data (117 patients without D-Dimer measurements in the princeps cohort of 289 patients) (Lines 80-82): “We identified consecutive COVID-19 patients admitted in nine general departments and with available D-Dimers measurements at Strasbourg University Hospital (two centers: Nouvel Hôpital Civil (NHC) and CHU Hautepierre) from February 25 to April 19, 2020 (Figure 1)”.
L81: with available D-Dimers measurements the authors should mention the method and discussion the importance of the method. The authors should refer to Hardy M et al. Thomb J. 2020 Sep 7;18:17 (where this point is addressed in detail). This point also explains why authors prefer to use ratios in comparison to the upper limit of the normal range.
“Therefore, their performance at higher values, such as those proposed for initiating high dose anticoagulation in COVID-19 patients (i.e., over 3000 ng/mL), is probably suboptimal and would need to be assessed in order to avoid the use of inaccurate results in these patients. For example, external quality controls performed with moderately elevated D-dimers samples (target value 4000 ng/mL FEU) identified method-specific D-dimers means ranging from 470 to 10,150 ng/mL FEU (all methods coefficient of variation: 23%) [68]. As a possible solution, the threshold values of plasma D-dimers could be adjusted based on assay methodology. For example, for DIC diagnosis according to the ISTH definition, the appropriate D-dimer cut-off value for 2 points ranged from 3500 ng/mL to 6500 ng/mL, depending on the reagents used [69].”
We have added the methodology used to measure D-Dimers in the “Methods” section of the main text (Line 102-104): “D-dimers were measured on STA-R® Evolution (Diagnostica Stago ®, Asnières-sur-Seine, France) with standard commercial reagents and protocols.”
We thank the Reviewer 2 for this relevant reference which we have added and discussed in the “Study Limitations” section (Lines 299-303): “Finally, while D-Dimers level may be an interesting biomarker to reflect the severity of lung injury during COVID-19, it may exist limitations of measurement of the plasma concentration of D-dimers depending on the methodology used and translated by a large inter-laboratory variability [39]. Tests performance seems to be suboptimal in particular for highest values. This point justifies the use of ratios in comparison to the upper limit of the normal range of D-Dimers.”
L90-92: “Thromboprophylaxis was achieved with Enoxaparin at 0.4 mL per day, Fondaparinux at 2.5 mg per day or unfractionated heparin at 200 IU per hour (standard preventive treatment) or with Enoxaparin at 0.4 mL twice per day (reinforced preventive treatment)”
For LMWH and UFH:please use IU/24h or IU/kg/24h
We have amended this part as follows (Lines 91-94): “When decided, thromboprophylaxis was achieved with Enoxaparin at 4000 international units (IU)/24 hours, Fondaparinux at 2.5 mg/24 hours or unfractionated heparin at 4800 IU/24 hours (standard preventive treatment), with Enoxaparin at 4000 IU twice /24 hours (reinforced preventive treatment) or with Enoxaparin 100 IU/kg twice /24 hours (therapeutic)”.
How many patients in each group?
We have added these findings in the Table 1.
L165 why abbreviations appear here?
“Abbreviations” is part of the Figure 1 caption: “Figure. 1 Flow-chart of the study. Among 289 COVID-19 patients admitted to general ward at Strasbourg Hospital, France between February 25 and April 19, 2020, this study included 123 patients after exclusion of 117 patients without available D-Dimers levels and 49 patients with venous thromboembolism during hospitalization. According to a D-Dimers staging following the CAHA classification, 67 patients (54.5%) were in Stage 1, 30 (24.4%) in Stage 2 and 26 (21.1%) in Stage 3.
Abbreviations: COVID-19, coronavirus disease 2019; ICU, intensive care unit; VTE, venous thromboembolism; ULN, upper limit of normal”.
In the discussion, the authors should discuss the limitations of D-dimers see previous reference
As previously mentioned, we have discussed this point in the “Study Limitations” section (299-303): “Finally, while D-Dimers level may be an interesting biomarker to reflect the severity of lung injury during COVID-19, it may exist limitations of measurement of the plasma concentration of D-dimers depending on the methodology used and translated by a large inter-laboratory variability [39]. Tests performance seems to be suboptimal in particular for highest values. This point justifies the use of ratios in comparison to the upper limit of the normal range in comparison with absolute values of D-Dimers.”